# The Upper and Lower Solution Method for a Class of Interval Boundary Value Problems

**Yanzong Yan** [1,*], **Zhiyong Xiao** [1] **and Zengtai Gong** [2,*]

[1] School of Mathematics and Statistics, Longdong University, Qingyang 745000, China; xiaozhiyong1983@163.com

[2] College of Mathematics and Statistics, Northwest Normal University, Lanzhou 730070, China

[*] Correspondence: ldxyyyz@163.com (Y.Y.); gongzt@nwnu.edu.cn (Z.G.)

**Abstract:** In this paper, the upper and lower solution method is proposed in order to solve the second order interval boundary value problem. We study first a class of linear interval boundary value problems and then investigate a class of nonlinear interval boundary value problems by the upper and lower solution method under the gH-derivative, and we prove that there exist at least two solutions.

**Keywords:** interval-valued functions; partial orders; interval boundary value problems; upper solution and lower solution method; gH-derivative

## 1. Introduction

In the process of mathematical modeling for solving problems, the initial data or parameter values are often uncertain due to measurement error. People often express these data and parameters as an interval number or fuzzy number. 1979, Markov proposed the interval-valued calculus [1]. This paper remained essentially un-cited for more than 30 years and was "rediscovered" after the publication of [2–4]. Stefanini considered a generalization of the Hukuhara difference and division for interval arithmetic and generalized Hukuhara differentiability of interval-valued functions and interval differential equations.

Recently, the interest for this topic increased significantly, in particular after the implementation of specific tools and classes in the C++ and Julia (among others) programming languages, or in computational systems, such as MATLAB or Mathematica [5]. The research activity in the calculus for interval-valued or set-valued functions is now very extended, particularly in connection with the more general calculus for fuzzy-valued functions with applications to almost all fields of applied mathematics [6–8].

Interval-valued differential equations are introduced as a good tool to study non-probabilistic uncertainty in real world phenomena. 2009, Stefanini and Bede studied several kinds of derivatives of an interval-valued function, and provided some properties of solutions to interval-valued differential equations under the gH-derivative [4]. 2011, Chalco-Cano et al. revisited the expression of the gH-derivative of an interval-valued function in terms of the endpoints functions [9]. In 2013, Lupulescu discussed the gH-differentiability of interval-valued functions, and studied interval differential equations on time-scales [10]. In 2017, by using a Krasnoselskii–Krein-type condition, Hoa, Lupulescu and O'Regan studied the existence and uniqueness of the solutions to initial value problems of fractional interval-valued differential equations [11].

In 2018, by applying the monotone iterative technique, Hoa considered the extremal solutions to initial value problems of fractional interval-valued integro-differential equations [12]. These studies expanded the scope of the research on interval-valued differential equations.

It is well known that the upper and lower solution method is a powerful tool for the solvability of differential equation [13]. Rodríguez-López applied the upper and lower solution method to develop a monotone iterative technique to approximate extremal solutions for the initial value problem relative to a fuzzy differential equation in a fuzzy functional interval [14]. Motivated by this idea, in order to solve the nonlinear interval boundary value problem

$$\begin{cases} U''(x) = F(x, u(x)), x \in I, \\ U(0) = A, \ U(1) = B, \end{cases}$$

where $A, B \in \mathcal{K}_C, U(x) \in C^2(I, \mathcal{K}_C), F(x, U) \in C(I \times \mathcal{K}_C, \mathcal{K}_C), I = [0, 1]$, we propose an upper and lower solution method and obtain at least four solutions similar to linear fuzzy boundary value problems.

In what follows, we introduce some preliminaries, in Section 3, we study a class of linear interval boundary value problems and give conditions that ensure that linear interval boundary value problems have solutions, and, in Section 4, we propose an upper and lower solution method for a class of nonlinear interval boundary value problems. In the last section, we give a example to illustrate the effectiveness of the results in this paper.

## 2. Preliminaries

In this section, we introduce some preliminaries that can be found in [7].

We denote by $\mathcal{K}_C$ the family of all bounded closed intervals in $\mathbb{R}$, i.e.,

$$\mathcal{K}_C = \{[a^-, a^+] | a^-, a^+ \in \mathbb{R} \text{ and } a^- \leqslant a^+\}.$$

The well-known midpoint-radius representation is very useful: for $A = [a^-, a^+]$, and we define the midpoints $\hat{a}$ and $\tilde{a}$, respectively, by

$$\hat{a} = \frac{a^- + a^+}{2} \text{ and } \tilde{a} = \frac{a^+ - a^-}{2},$$

so that $a^- = \hat{a} - \tilde{a}$ and $a^+ = \hat{a} + \tilde{a}$. We will denote the interval by $A = [a^-, a^+]$ or, in midpoint notation, by $A = (\hat{a}; \tilde{a})$; thus,

$$\mathcal{K}_C = \{(\hat{a}; \tilde{a}) | \hat{a}; \tilde{a} \in \mathbb{R} \text{ and } \tilde{a} \geqslant 0\}.$$

The gH-difference of two intervals always exists and, in midpoint notation, is given by

$$A \ominus_{gH} B = (\hat{a} - \hat{b}; |\tilde{a} - \tilde{b}|);$$

the gH-addition for intervals is defined by

$$A \oplus_{gH} B = A \ominus_{gH} (-B) = (\hat{a} + \hat{b}; |\tilde{a} - \tilde{b}|).$$

Endowed with the Pompeiu–Hausdorff distance $d_H : \mathcal{K}_C \times \mathcal{K}_C \to \mathbb{R}_+ \cup \{0\}$, defined by

$$d_H(A, B) = \max \left\{ \max_{a \in A} d(a, B), \max_{b \in B} d(b, A) \right\}$$

with $d(a, B) = \min_{b \in B} |a - b|$ and given also as $d_H(A, B) = \|A \ominus_{gH} B\|$ (here, for $C \in \mathcal{K}_C, \|C\| = \max\{|c|; c \in C\} = d_H(C, \{0\}))$, the metric space $(\mathcal{K}_C, d_H)$ is complete.

**Definition 1.** (*[7]*) *Given two intervals $A = [a^-, a^+] = (\hat{a}; \tilde{a})$ and $B = [b^-, b^+] = (\hat{b}; \tilde{b})$ and $\gamma^- \leqslant 0, \gamma^+ \geqslant 0$ (eventually $\gamma^- = -\infty$ and/or $\gamma^+ = +\infty$), we define the following order relation, denoted $\precapprox_{\gamma^-, \gamma^+}$,*

$$A \precapprox_{\gamma^-, \gamma^+} B \iff \begin{cases} \hat{a} \leqslant \hat{b}, \\ \tilde{a} \geqslant \tilde{b} + \gamma^+(\hat{a} - \hat{b}), \\ \tilde{a} \leqslant \tilde{b} + \gamma^-(\hat{a} - \hat{b}). \end{cases}$$

The space $(\mathcal{K}_C, \precapprox_{\gamma^-, \gamma^+})$ is a lattice. The reverse order is defined by $A \succapprox_{\gamma^-, \gamma^+} B \iff B \precapprox_{\gamma^-, \gamma^+} A$, i.e.,

$$A \succapprox_{\gamma^-, \gamma^+} B \iff \begin{cases} \hat{a} \geqslant \hat{b}, \\ \tilde{a} \leqslant \tilde{b} + \gamma^+(\hat{a} - \hat{b}), \\ \tilde{a} \geqslant \tilde{b} + \gamma^-(\hat{a} - \hat{b}). \end{cases}$$

An interval-valued function is defined to be any $F : [a, b] \longrightarrow \mathcal{K}_C$ with $F(x) = [f^-(x), f^+(x)] \in \mathcal{K}_C$ and $f^-(x) \leqslant f^+(x)$ for all $x \in [a, b]$. In midpoint representation, we write $F(x) = (\hat{f}(x); \tilde{f}(x))$, where $\hat{f}(x) \in \mathbb{R}$ is the midpoint value of interval $F(x)$ and $\tilde{f}(x) \in \mathbb{R}^+ \cup \{0\}$ is the nonnegative half-length of $F(x)$ :

$$\hat{f}(x) = \frac{f^+(x) + f^-(x)}{2} \text{ and } \tilde{f}(x) = \frac{f^+(x) - f^-(x)}{2} \geqslant 0,$$

so that

$$f^-(x) = \hat{f}(x) - \tilde{f}(x) \text{ and } f^+(x) = \hat{f}(x) + \tilde{f}(x).$$

Limits and continuity can be characterized, in the Pompeiu–Hausdorff metric $d_H$ for intervals, by the gH-difference. For a function $F : K \longrightarrow \mathcal{K}_C, K \in \mathbb{R}$, an interval $L = [l^-, l^+] \in \mathcal{K}_C$ and an accumulation point $x_0$, we have

$$\lim_{x \to x_0} F(x) = L \iff \lim_{x \to x_0} (F(x) \ominus_{gH} L) = 0,$$

where the limits are in the metric $d_H$. If, in addition, $x_0 \in K$, we have

$$\lim_{x \to x_0} F(x) = F(x_0) \iff \lim_{x \to x_0} (F(x) \ominus_{gH} F(x_0)) = 0.$$

In midpoint notation, let $F(x) = (\hat{f}(x); \tilde{f}(x))$ and $L = (\hat{l}; \tilde{l})$; then, the limits and continuity can be expressed, respectively, as

$$\lim_{x \to x_0} F(x) = L \iff \lim_{x \to x_0} \hat{f}(x) = \hat{l} \text{ and } \lim_{x \to x_0} \tilde{f}(x) = \tilde{l}$$

and

$$\lim_{x \to x_0} F(x) = F(x_0) \iff \lim_{x \to x_0} \hat{f}(x) = \hat{f}(x_0) \text{ and } \lim_{x \to x_0} \tilde{f}(x) = \tilde{f}(x_0).$$

Let $C(I, \mathcal{K}_C)$ be the set of all continuous interval-value functions.

**Theorem 1.** *Let $C_n, A, B \in \mathcal{K}_C, n = 1, 2, \cdots$, and $A \succapprox_{\gamma^-, \gamma^+} C_n \precapprox_{\gamma^-, \gamma^+} B$. Then, there exists an interval $C \in \mathcal{K}_C$ such that*

$$\lim_{n \to \infty} C_n = C.$$

**Proof.** Let $C_n = (\hat{c}_n; \tilde{c}_n), A = (\hat{a}; \tilde{a}), B = (\hat{b}; \tilde{b})$. Since

$$A \succapprox_{\gamma^-, \gamma^+} C_n \precapprox_{\gamma^-, \gamma^+} B,$$

we have

$$\begin{cases} \hat{a} \leqslant \hat{c}_n, \\ \tilde{a} \geqslant \tilde{c}_n + \gamma^+(\hat{a} - \hat{c}_n), \\ \tilde{a} \leqslant \tilde{c}_n + \gamma^-(\hat{a} - \hat{c}_n) \end{cases} \quad \text{and} \quad \begin{cases} \hat{c}_n \leqslant \hat{b}, \\ \tilde{c}_n \geqslant \tilde{b} + \gamma^+(\hat{c}_n - \hat{b}), \\ \tilde{c}_n \leqslant \tilde{b} + \gamma^-(\hat{c}_n - \hat{b}). \end{cases}$$

It follows that

$$\hat{a} \leqslant \hat{c}_n \leqslant \hat{b},$$

$$\tilde{a} - \gamma^+\hat{a} + \gamma^+\hat{b} \geqslant \tilde{a} - \gamma^+\hat{a} + \gamma^+\hat{c}_n \geqslant \tilde{c}_n \geqslant \tilde{b} + \gamma^+(\hat{c}_n - \hat{b}) \geqslant \tilde{c}_n \geqslant \tilde{b} + \gamma^+(\hat{a} - \hat{b})$$

when $\gamma^+ < +\infty$,

$$\tilde{a} - \gamma^-\hat{a} + \gamma^-\hat{b} \leqslant \tilde{a} - \gamma^-\hat{a} + \gamma^-\hat{c}_n \leqslant \tilde{c}_n \leqslant \tilde{b} + \gamma^-\hat{c}_n - \gamma^-\hat{b} \leqslant \tilde{b} + \gamma^-\hat{a} - \gamma^-\hat{b}$$

when $\gamma^- > -\infty$. Hence, there exist $\hat{c}$ and $\tilde{c}$ such that

$$\lim_{n \to \infty} \hat{c}_n = \hat{c}, \quad \lim_{n \to \infty} \tilde{c}_n = \tilde{c}.$$

Let $C = (\hat{c}; \tilde{c})$. It is clear that

$$\lim_{n \to \infty} C_n = C.$$

□

**Theorem 2.** *Let* $F_n(x) = (\hat{f}_n(x); \tilde{f}_n(x)), F(x) = (\hat{f}(x); \tilde{f}(x))$ *be interval-value functions. Then,*

$$\lim_{n \to \infty} F_n(x) = F(x)$$

*if and only if*

$$\lim_{n \to \infty} \hat{f}_n(x) = \hat{f}(x), \quad \lim_{n \to \infty} \tilde{f}_n(x) = \tilde{f}(x).$$

**Proof.** It is similar to prove that

$$\lim_{x \to x_0} F(x) = L \iff \lim_{x \to x_0} \hat{f}(x) = \hat{l} \text{ and } \lim_{x \to x_0} \tilde{f}(x) = \tilde{l}.$$

□

**Definition 2.** *Given two interval-value functions $F(x)$ and $G(x)$, we define the distance of $F(x)$ and $G(x)$ as*

$$D_H(F, G) = \max_{x \in [0,1]} d_H(F(x), G(x)).$$

**Theorem 3.** *Let $F_n(x), E(x), G(x) \in C(I, \mathcal{K}_C)$ be interval-value functions and*

$$E(x) \precsim_{\gamma^-, \gamma^+} F_n(x) \precsim_{\gamma^-, \gamma^+} G(x).$$

*If*

$$\lim_{n \to \infty} F_n(x) = F(x),$$

*then*

$$\lim_{n \to \infty} F_n = F.$$

**Proof.** Let $F_n(x) = (\hat{f}_n(x); \tilde{f}_n(x)), E(x) = (\hat{e}(x); \tilde{e}(x)), G(x) = (\hat{g}(x); \tilde{g}(x))$. It follows that $\hat{f}_n(x)$ and $\tilde{f}_n(x)$ are two continuous functions. Similar to the proof of Theorem 1, it is easy obtain that $\hat{f}_n(x)$ and $\tilde{f}_n(x)$ are uniformly bounded. Hence,

$$\lim_{n \to \infty} \hat{f}_n = \hat{f}, \quad \lim_{n \to \infty} \tilde{f}_n = \tilde{f}.$$

Therefore,

$$\lim_{n \to \infty} F_n = F.$$

□

**Definition 3.** *Let $x_0 \in (a, b)$ and $h$ be such that $x_0 + h \in (a, b)$, and then the gH-derivative of a function $F : (a, b) \to \mathcal{K}_C$ at $x_0$ is defined as*

$$F'_{gH}(x_0) = \lim_{h \to 0} \frac{1}{h} [F(x_0 + h) \ominus_{gH} F(x_0)]$$

*if the limit exists. The interval $F'_{gH}(x_0) \in \mathcal{K}_C$ is called the generalized Hukuhara derivative of $F$ (gH-derivative for short) at $x_0$.*

For a gH-differentiable function, higher-order gH-derivatives are defined analogously to the ordinary case using the gH-differences applied to the gH-derivatives of previous order.

**Definition 4.** *Let $F : (a, b) \to \mathcal{K}_C$ be gH-differentiable on $(a, b)$ and $x_0 \in (a, b)$ and $h$ be such that $x_0 + h \in (a, b)$. The second order gH-derivative of $F(x)$ at $x_0$ is defined as*

$$F''_{gH}(x_0) = \lim_{h \to 0} \frac{1}{h} [F'_{gH}(x_0 + h) \ominus_{gH} F'_{gH}(x_0)]$$

*if the limit exists. The interval $F''_{gH}(x_0) \in \mathcal{K}_C$ is called the second order gH-derivative of $F$ at $x_0$.*

**Remark 1.** *In the case of an interval-valued function in the form $F(x) = (\hat{f}(x); \tilde{f}(x))$ with $\hat{f}(x) = |\varphi(x)|$ where $\hat{f}(x)$ and $\varphi(x)$ have derivatives $\hat{f}^{(i)}(x)$, $\varphi^{(i)}(x)$ for $i = 1, 2$, we have that $F(x)$ has all the gH-derivatives $F^i_{gH}(x) = (\hat{f}^{(i)}(x); |\varphi^{(i)}(x)|), i = 1, 2$. Let $C^i(I, \mathcal{K}_C)$ be the set of all $i$ order continuous differentiable interval-value functions, $i = 1, 2$.*

### 3. The Linear Interval Boundary Problems

In this section, we consider a class of linear interval boundary problems under the gH-derivative. Let

$$\begin{cases} U''(t) = F(t), \ t \in I, \\ U(0) = A, \ U(1) = B, \end{cases} \tag{1}$$

where $A, B \in \mathcal{K}_C, F(t) \in C(I, \mathcal{K}_C)$.

**Definition 5.** *If $U(t) \in C^2(I, \mathcal{K}_C)$ and $U(t)$ satisfies Problem (1), then we say that $U(t)$ is a solution of Problem (1).*

The following theorems concern the existence of solutions of a two-point boundary value problem of linear interval differential equation under the gH-derivative.

**Theorem 4.** *Let $F(t) = (\hat{f}(x); \tilde{f}(x)) \in C(I, \mathcal{K}_C), A = (\hat{a}, \tilde{a}), B = (\hat{b}, \tilde{b})$. Then, Problem (1) has at least four solutions*

$$U(t) = (\hat{u}(x); \tilde{u}(x)),$$

*where*

$$\hat{u}(x) = \int_0^x \int_0^t \hat{f}(t) \mathrm{d}t \mathrm{d}x + \left( \hat{b} - \hat{a} - \int_0^1 \int_0^t \hat{f}(t) \mathrm{d}t \mathrm{d}x \right) x + \hat{a},$$

$$\tilde{u}(x) = \left| \int_0^x \int_0^t (\pm \tilde{f}(t)) \mathrm{d}t \mathrm{d}x + \left[ \hat{b} - \hat{a} - \int_0^1 \int_0^t (\pm \tilde{f}(t)) \mathrm{d}t \mathrm{d}x \right] x + \hat{a} \right|$$

*or*

$$\tilde{u}(x) = \left| \int_0^x \int_0^t (\pm \tilde{f}(t)) \mathrm{d}t \mathrm{d}x + \left[ -\hat{b} - \hat{a} - \int_0^1 \int_0^t (\pm \tilde{f}(t)) \mathrm{d}t \mathrm{d}x \right] x + \hat{a} \right|$$

*or*

$$\tilde{u}(x) = \left| \int_0^x \int_0^t \left( \pm \tilde{f}(t) \right) dt dx + \left[ \hat{b} + \hat{a} - \int_0^1 \int_0^t \left( \pm \tilde{f}(t) \right) dt dx \right] x - \hat{a} \right|$$

*or*

$$\tilde{u}(x) = \left| \int_0^x \int_0^t \left( \pm \tilde{f}(t) \right) dt dx + \left[ -\hat{b} + \hat{a} - \int_0^1 \int_0^t \left( \pm \tilde{f}(t) \right) dt dx \right] x - \hat{a} \right|$$

**Proof.** Let $U(t) = (\hat{u}(x); \tilde{u}(x))$ is a solution of Problem (1), where $\tilde{u}(x) = |\varphi(x)|$. By Definition 4 and Remark 1, Problem (1) is equivalent to the following problems:

$$\begin{cases} \hat{u}''(x) = \hat{f}(x), \\ \hat{u}(0) = \hat{a}, \\ \hat{u}(1) = \hat{b}, \end{cases} \quad \text{and} \quad \begin{cases} |\varphi''(x)| = \tilde{f}(x), \\ |\varphi(0)| = \tilde{a}, \\ |\varphi(1)| = \tilde{b}. \end{cases} \tag{2}$$

From (2), we have

$$\hat{u}'(x) = \int_0^x \hat{f}(x) dx + c_1,$$

$$\hat{u}(x) = \int_0^x \int_0^t \hat{f}(t) dt dx + c_1 x + c_2.$$

$$\varphi'(x) = \int_0^x \left( \pm \tilde{f}(x) \right) dx + c_3,$$

$$\varphi(x) = \int_0^x \int_0^t \left( \pm \tilde{f}(t) \right) dt dx + c_1 x + c_4.$$

By $\hat{u}(0) = \hat{a}$ and $\hat{u}(1) = \hat{b}$, it is easy to find that

$$c_1 = \hat{b} - \hat{a} - \int_0^1 \int_0^t \hat{f}(t) dt dx \quad \text{and} \quad c_2 = \hat{a},$$

that is to say

$$\hat{u}(x) = \int_0^x \int_0^t \hat{f}(t) dt dx + \left( \hat{b} - \hat{a} - \int_0^1 \int_0^t \hat{f}(t) dt dx \right) x + \hat{a}.$$

By $|\varphi(0)| = \tilde{a}$ and $|\varphi(1)| = \tilde{b}$, we have $\varphi(0) = \pm\tilde{a}$ and $\varphi(1) = \pm\tilde{b}$. If $\varphi(0) = \tilde{a}, \varphi(1) = \tilde{b}$, then

$$\varphi(x) = \int_0^x \int_0^t \left( \pm \tilde{f}(t) \right) dt dx + \left[ \hat{b} - \hat{a} - \int_0^1 \int_0^t \left( \pm \tilde{f}(t) \right) dt dx \right] x + \hat{a}.$$

If $\varphi(0) = \tilde{a}, \varphi(1) = -\tilde{b}$, then

$$\varphi(x) = \int_0^x \int_0^t \left( \pm \tilde{f}(t) \right) dt dx + \left[ -\hat{b} - \hat{a} - \int_0^1 \int_0^t \left( \pm \tilde{f}(t) \right) dt dx \right] x + \hat{a}.$$

If $\varphi(0) = -\tilde{a}, \varphi(1) = \tilde{b}$, then

$$\varphi(x) = \int_0^x \int_0^t \left( \pm \tilde{f}(t) \right) dt dx + \left[ \hat{b} + \hat{a} - \int_0^1 \int_0^t \left( \pm \tilde{f}(t) \right) dt dx \right] x - \hat{a}.$$

If $\varphi(0) = -\tilde{a}, \varphi(1) = -\tilde{b}$, then

$$\varphi(x) = \int_0^x \int_0^t \left( \pm \tilde{f}(t) \right) dt dx + \left[ -\hat{b} + \hat{a} - \int_0^1 \int_0^t \left( \pm \tilde{f}(t) \right) dt dx \right] x - \hat{a}.$$

Therefore, Problem (1) has at least four solutions

$$U(t) = (\hat{u}(x); \tilde{u}(x)),$$

where

$$\hat{u}(x) = \int_0^x \int_0^t \hat{f}(t)\mathrm{d}t\mathrm{d}x + \left( \hat{b} - \hat{a} - \int_0^1 \int_0^t \hat{f}(t)\mathrm{d}t\mathrm{d}x \right) x + \hat{a},$$

$$\tilde{u}(x) = \left| \int_0^x \int_0^t \left( \pm \tilde{f}(t) \right)\mathrm{d}t\mathrm{d}x + \left[ \hat{b} - \hat{a} - \int_0^1 \int_0^t \left( \pm \tilde{f}(t) \right)\mathrm{d}t\mathrm{d}x \right] x + \hat{a} \right|$$

or

$$\tilde{u}(x) = \left| \int_0^x \int_0^t \left( \pm \tilde{f}(t) \right)\mathrm{d}t\mathrm{d}x + \left[ -\hat{b} - \hat{a} - \int_0^1 \int_0^t \left( \pm \tilde{f}(t) \right)\mathrm{d}t\mathrm{d}x \right] x + \hat{a} \right|$$

or

$$\tilde{u}(x) = \left| \int_0^x \int_0^t \left( \pm \tilde{f}(t) \right)\mathrm{d}t\mathrm{d}x + \left[ \hat{b} + \hat{a} - \int_0^1 \int_0^t \left( \pm \tilde{f}(t) \right)\mathrm{d}t\mathrm{d}x \right] x - \hat{a} \right|$$

or

$$\tilde{u}(x) = \left| \int_0^x \int_0^t \left( \pm \tilde{f}(t) \right)\mathrm{d}t\mathrm{d}x + \left[ -\hat{b} + \hat{a} - \int_0^1 \int_0^t \left( \pm \tilde{f}(t) \right)\mathrm{d}t\mathrm{d}x \right] x - \hat{a} \right|$$

$\square$

**Theorem 5.** *Let $U(x)$ be a solution of Problem (1), and define operator $T$ as*

$$T : F(x) \rightarrow U.$$

*Then, operator $T$ is a continuous operator.*

**Proof.** For convenience, assume a solution of Problem (1) is

$$U(t) = (\hat{u}(x); \tilde{u}(x)),$$

where

$$\hat{u}(x) = \int_0^x \int_0^t \hat{f}(t)\mathrm{d}t\mathrm{d}x + \left( \hat{b} - \hat{a} - \int_0^1 \int_0^t \hat{f}(t)\mathrm{d}t\mathrm{d}x \right) x + \hat{a},$$

$$\tilde{u}(x) = \left| \int_0^x \int_0^t \left( \tilde{f}(t) \right)\mathrm{d}t\mathrm{d}x + \left[ \hat{b} - \hat{a} - \int_0^1 \int_0^t \left( \tilde{f}(t) \right)\mathrm{d}t\mathrm{d}x \right] x + \hat{a} \right|.$$

It is clear that operator $T$ is a continuous operator. $\square$

### 4. The Nonlinear Interval Boundary Value Problems

In what follows, we consider a class of nonlinear interval boundary problems

$$\begin{cases} U''(x) = F(x, U(x)), \ x \in I, \\ U(0) = A, \ U(1) = B. \end{cases} \tag{3}$$

where $A, B \in \mathcal{K}_C, F(t, U) \in C(I \times \mathcal{K}_C, \mathcal{K}_C)$.

**Definition 6.** *If $U(x) \in C^2(I, \mathcal{K}_C)$ and $U(x)$ satisfy Problem (3), then we say that $U(x)$ is a solution of Problem (3).*

We present the concept of upper and lower solution of a two-point boundary value problem of nonlinear interval differential equation under the gH-derivative in the following definition.

**Definition 7.** $\overline{U}(t) \in C^2\big(I, \mathcal{K}_C\big)$ *is said to be an upper solution of Problem* (3) *if*

$$\begin{cases} \overline{U}''(x) \precapprox_{\gamma^-,\gamma^+} F\big(x,\overline{U}(x)\big),\ x \in I, \\ \overline{U}(0) \succapprox_{\gamma^-,\gamma^+} A,\ \overline{U}(1) \succapprox_{\gamma^-,\gamma^+} B, \end{cases}$$

$\underline{U}(t) \in C^2\big(I, \mathcal{K}_C\big)$ *is said to be a lower solution of Problem* (3) *if*

$$\begin{cases} \underline{U}''(x) \succapprox_{\gamma^-,\gamma^+} F\big(x,\underline{U}(x)\big),\ x \in I, \\ \underline{U}(0) \precapprox_{\gamma^-,\gamma^+} A,\ \underline{U}(1) \precapprox_{\gamma^-,\gamma^+} B. \end{cases}$$

$U(t)$ *is said to be a solution of Problem* (3) *if* $U(t)$ *is an upper solution and is also a lower solution of Problem* (3).

The following theorems concern the existence of solutions of a two-point boundary value problem of a nonlinear interval differential equation under the gH-derivative.

**Theorem 6.** *Let* $\overline{U}(x)$, $\underline{U}(x)$ *be an upper solution and a lower solution of Problem* (3), *and* $\underline{U}(t) \precapprox_{\gamma^-,\gamma^+} \overline{U}(t)$. *If* $F(t,U) \in C\big(I \times \mathcal{K}_C, \mathcal{K}_C\big)$, *and* $F(x,V) \succapprox_{\gamma^-,\gamma^+} F(x,W)$ *when* $V \precapprox_{\gamma^-,\gamma^+} W$, *then Problem* (3) *exists at least two solutions.*

**Proof.** Since $\underline{U}(x)$ is a lower solution of Problem (3), then

$$\begin{cases} \underline{U}''(x) \succapprox_{\gamma^-,\gamma^+} F\big(x,\underline{U}(x)\big),\ x \in I, \\ \underline{U}(0) \precapprox_{\gamma^-,\gamma^+} A,\ \underline{U}(1) \precapprox_{\gamma^-,\gamma^+} B. \end{cases}$$

By Theorem 4, we know that there exists $V_1(x) \in C^2\big(I, \mathcal{K}_C\big)$, which is the solution of the linear interval boundary value problem

$$\begin{cases} U''(t) = F\big(t,\underline{U}(t)\big),\ x \in I, \\ U(0) = A,\ U(1) = B. \end{cases}$$

Hence,

$$\begin{cases} \underline{U}''(x) \succapprox_{\gamma^-,\gamma^+} V_1''(x), \\ \underline{U}(0) \precapprox_{\gamma^-,\gamma^+} V_1(0),\ \underline{U}(1) \precapprox_{\gamma^-,\gamma^+} V_1(1). \end{cases}$$

It follows that

$$\underline{\hat{U}}''(x) \geqslant \hat{V}_1''(x),$$

$$\gamma^+ \underline{\hat{U}}''(x) - \underline{\tilde{U}}''(x) \geqslant \gamma^+ \hat{V}_1''(x) - \tilde{V}_1''(x),$$

$$-\gamma^- \underline{\hat{U}}''(x) + \underline{\tilde{U}}''(x) \geqslant -\gamma^- \hat{V}_1''(x) + \tilde{V}_1''(x),$$

$$\underline{\hat{U}}(0) \leqslant \hat{V}_1(0),$$

$$\gamma^+ \underline{\hat{U}}(0) - \underline{\tilde{U}}(0) \leqslant \gamma^+ \hat{V}_1(0) - \tilde{V}_1(0),$$

$$-\gamma^- \underline{\hat{U}}(0) + \underline{\tilde{U}}(0) \leqslant -\gamma^- \hat{V}_1(0) + \tilde{V}_1(0),$$

$$\underline{\hat{U}}(1) \leqslant \hat{V}_1(1),$$

$$\gamma^+ \underline{\hat{U}}(1) - \underline{\tilde{U}}(1) \leqslant \gamma^+ \hat{V}_1(1) - \tilde{V}_1(1),$$

$$-\gamma^- \underline{\hat{U}}(1) + \underline{\tilde{U}}(1) \leqslant -\gamma^- \hat{V}_1(1) + \tilde{V}_1(1).$$

Since
$$\underline{\hat{U}}''(x) \geqslant \hat{V}_1''(x),$$
$$\underline{\hat{U}}(0) \leqslant \hat{V}_1(0),$$
$$\underline{\hat{U}}(1) \leqslant \hat{V}_1(1),$$

we obtain
$$\underline{\hat{U}}(x) \leqslant \hat{V}_1(x).$$

Similarly, we also find

$$\gamma^+ \underline{\hat{U}}(x) - \underline{\tilde{U}}(x) \leqslant \gamma^+ \hat{V}_1(x) - \tilde{V}_1(x),$$

$$-\gamma^- \underline{\hat{U}}(x) + \underline{\tilde{U}}(x) \leqslant -\gamma^- \hat{V}_1(x) + \tilde{V}_1(x),$$

i.e.,

$$\begin{cases} \underline{\hat{U}}(x) \leqslant \hat{V}_1(x), \\ \gamma^+ \underline{\hat{U}}(x) - \underline{\tilde{U}}(x) \leqslant \gamma^+ \hat{V}_1(x) - \tilde{V}_1(x), \\ -\gamma^- \underline{\hat{U}}(x) + \underline{\tilde{U}}(x) \leqslant -\gamma^- \hat{V}_1(x) + \tilde{V}_1(x). \end{cases}$$

Then,
$$\underline{U}(x) \precsim_{\gamma^-,\gamma^+} V_1(x).$$

By similar reasoning, if $\overline{U}$ is an upper solution of Problem (3), and $U^1$ is a solution of the linear fuzzy boundary value problem

$$\begin{cases} U''(x) = F(x, \overline{U}(x)), \ x \in I, \\ U(0) = A, \ U(1) = B. \end{cases}$$

We find
$$\overline{U} \succsim_{\gamma^-,\gamma^+} U^1.$$

Assume $\underline{U} \precsim_{\gamma^-,\gamma^+} V \precsim_{\gamma^-,\gamma^+} W \precsim_{\gamma^-,\gamma^+} \overline{U}$, let $TV$ be a solution of the linear interval boundary value problem

$$\begin{cases} U''(x) = F(x, V(x)), \ x \in I, \\ U(0) = A, \ U(1) = B. \end{cases}$$

Since,
$$F(x, V) \succsim_{\gamma^-,\gamma^+} F(x, W),$$

then
$$(TV)'' \succsim_{\gamma^-,\gamma^+} (TW)'', \quad (TV)(0) = (TW)(0), \quad (TV)(1) = (TW)(1).$$

Hence,
$$T\hat{V}''(x) \geqslant T\hat{W}''(x),$$

$$\gamma^+ T\hat{V}''(x) - T\tilde{V}''(x) \geqslant \gamma^+ T\hat{W}''(x) - T\tilde{W}''(x),$$

$$-\gamma^- T\hat{V}''(x) + T\tilde{V}''(x) \geqslant -\gamma^- T\hat{W}''(x) + T\tilde{W}''(x),$$

$$T\hat{V}''(0) = T\hat{W}''(0),$$

$$\gamma^+ T\hat{V}''(0) - T\tilde{V}''(0) = \gamma^+ T\hat{W}''(0) - T\tilde{W}''(0),$$

$$-\gamma^- T\hat{V}''(0) + T\tilde{V}''(0) = -\gamma^- T\hat{W}''(0) + T\tilde{W}''(0),$$

$$T\hat{V}''(1) = T\hat{W}''(1),$$

$$\gamma^+ T\hat{V}''(1) - T\tilde{V}''(1) = \gamma^+ T\hat{W}''(1) - T\tilde{W}''(10),$$

$$-\gamma^- T\hat{V}''(1) + T\tilde{V}''(1) = -\gamma^- T\hat{W}''(1) + T\tilde{W}''(1).$$

It follows that

$$\begin{cases} T\hat{V}(x) \leqslant T\hat{W}(x), \\ \gamma^+ T\hat{V}(x) - T\tilde{V}(x) \leqslant \gamma^+ T\hat{W}(x) - T\tilde{W}(x), \\ -\gamma^- T\hat{V}(x) + T\tilde{W}V(x) \leqslant -\gamma^- T\hat{W}(x) + T\tilde{W}(x), \end{cases}$$

i.e.,

$$TV \precsim_{\gamma^-,\gamma^+} TW.$$

Therefore, $T$ is a monotone operator.
Since,

$$\underline{U}(x) \precsim_{\gamma^-,\gamma^+} V_1(x),$$

we obtain

$$V_1 = T\underline{U} \precsim_{\gamma^-,\gamma^+} TV_1 = V_2,$$

$$V_2 = TV_1 \precsim_{\gamma^-,\gamma^+} TV_2 = V_3,$$

and let $V_n = TV_{n-1}$, we have

$$V_0 = \underline{U} \precsim_{\gamma^-,\gamma^+} V_1 \precsim_{\gamma^-,\gamma^+} V_2 \precsim_{\gamma^-,\gamma^+} \cdots \precsim_{\gamma^-,\gamma^+} V_{n-1} \precsim_{\gamma^-,\gamma^+} V_n \precsim_{\gamma^-,\gamma^+} \cdots.$$

A similar argument for $U^n = TU^{n-1}$, we have

$$\cdots \precsim_{\gamma^-,\gamma^+} U^n \precsim_{\gamma^-,\gamma^+} U^{n-1} \precsim_{\gamma^-,\gamma^+} \cdots \precsim_{\gamma^-,\gamma^+} U^2 \precsim_{\gamma^-,\gamma^+} U^1 \precsim_{\gamma^-,\gamma^+} \overline{U} = U^0.$$

In addition, from

$$\underline{U} \precsim_{\gamma^-,\gamma^+} \overline{U}$$

we conclude that

$$V_1 = T\underline{U} \precsim_{\gamma^-,\gamma^+} T\overline{U} = U^1,$$

and

$$V_2 = TV_1 \precsim_{\gamma^-,\gamma^+} TU^1 = U^2.$$

Similarly,

$$V_n = TV_{n-1} \precsim_{\gamma^-,\gamma^+} TU^{n-1} = U^n.$$

In conclusion,

$$\underline{U} \precsim_{\gamma^-,\gamma^+} V_1 \precsim_{\gamma^-,\gamma^+} V_2 \precsim_{\gamma^-,\gamma^+} \cdots \precsim_{\gamma^-,\gamma^+} V_n \precsim_{\gamma^-,\gamma^+} \cdots$$

$$\precsim_{\gamma^-,\gamma^+} U^n \precsim_{\gamma^-,\gamma^+} \cdots \precsim_{\gamma^-,\gamma^+} U^2 \precsim_{\gamma^-,\gamma^+} U^1 \precsim_{\gamma^-,\gamma^+} \overline{U}.$$

By Theorem 1, there exist two interval-value functions $V(x), U(x) \in \mathcal{K}_C$ such that

$$\lim_{n\to\infty} V_n(x) = V(x), \quad \lim_{n\to\infty} U^n(x) = U(x),$$

and

$$\underline{V} \precsim_{\gamma^-,\gamma^+} V \precsim_{\gamma^-,\gamma^+} U \precsim_{\gamma^-,\gamma^+} \overline{U}.$$

By Theorems 3 and 5, we have

$$TV = V, \quad TU = U,$$

i.e., Problem (3) exists at least two solutions. $\square$

## 5. Example

In this section, we will give a example to illustrate the effectiveness of the results.

**Example 1.** *If $\gamma^- = -1, \gamma^+ = 1, U(x) = [u^-(x), u^+(x)]$, let*

$$\varphi(U) = \max_{0 \le x \le 1} \max \left\{ |u^-(x)|, |u^+(x)| \right\},$$

$$F(x, U) = a(x) \left( \frac{1}{1 + \varphi(u)} \right) C, \ (x, U) \in I \times \mathcal{K}_C.$$

*where*

$$a(t) \in C[0,1], \ 0 \le a(t) \le 1,$$

$$C = [c^-, c^+], \ c^- \ge 0.$$

*It is easy to check that $F(t, U) \in C(I \times \mathcal{K}_C, \mathcal{K}_C)$, and $F(x, V) \succsim_{\gamma^-, \gamma^+} F(x, W)$ when $V \precsim_{\gamma^-, \gamma^+} W$ if $v^-(x), w^-(x) \ge 0$.*

*Clearly, $\underline{U}(x) = x^2 C = [x^2 c^-, x^2 c^+]$ is a lower solution of Problem (3), when $\underline{U}(0) = 0 \precsim_{\gamma^-, \gamma^+} A, \underline{U}(1) = [c^-, c^+] \precsim_{\gamma^-, \gamma^+} B$.*

*$\overline{U}(x) = MC = [Mc^-, Mc^+]$ is an upper solution of Problem (3), when $\overline{U}(0) = [Mc^-, Mc^+] \succsim_{\gamma^-, \gamma^+} A, \overline{U}(1) = [Mc^-, Mc^+] \succsim_{\gamma^-, \gamma^+} B$.*

*Therefore, in this case, the conclusion of Theorem 6 holds.*

**Remark 2.** *$\underline{U}(x) = x^2 C = [x^2 c^-, x^2 c^+]$ is a lower solution of Problem (3). In fact, since*

$$\underline{U}(x) = x^2 C = [x^2 c^-, x^2 c^+] = \left( \frac{x^2(c^- + c^+)}{2}; \frac{x^2(c^+ - c^-)}{2} \right),$$

*we have*

$$\underline{U}'(x) = \left( x(c^- + c^+); x(c^+ - c^-) \right),$$

$$\underline{U}''(x) = \left( c^- + c^+; c^+ - c^- \right) = [2c^-, 2c^+] \succsim_{\gamma^-, \gamma^+} F(x, \underline{U}(x)).$$

*$\overline{U}(x) = MC = [Mc^-, Mc^+]$ is an upper solution of Problem (3). In fact, since*

$$\overline{U}(x) = MC = [Mc^-, Mc^+] = \left( \frac{M(c^- + c^+)}{2}; \frac{M(c^+ - c^-)}{2} \right),$$

*we have*

$$\overline{U}'(x) = (0; 0),$$

$$\overline{U}''(x) = (0; 0) = [0, 0] \precsim_{\gamma^-, \gamma^+} F(x, \underline{U}(x)).$$

## 6. Conclusions

In this paper, we studied a class of linear interval boundary value problems and then investigated a class of nonlinear interval boundary value problems by the upper and lower solution method under the gH-derivative. We found that there are at least four solutions for linear interval boundary value problems and at least two solutions for nonlinear interval boundary value problems. In our next works, we will consider the interval boundary problem when $F(x, U)$ is increasing for $U$.

**Author Contributions:** Conceptualization, Y.Y. and Z.G.; investigation, Z.X. All authors have read and agreed to the published version of the manuscript.

**Funding:** This research was funded by National Natural Science Foundation of China (Grant Nos. 12061067 and 61763044).

**Data Availability Statement:** Not applicable.

**Acknowledgments:** The authors would like to thank the referees for providing very helpful comments and suggestions.

**Conflicts of Interest:** The authors declare no conflict of interest.

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
