# Peer review of "The Upper and Lower Solution Method for a Class of Interval Boundary Value Problems"

_axioms, doi:10.3390/axioms10040269_

Round 1

Reviewer 1 Report

  1. The English in the present manuscript is not of publication quality and require major improvement. Please carefully proof-read spell check to eliminate grammatical errors.
  2. How the problem 3.1 has 4 solutions using theorem 3.2?
  3. how the definition 4.1 has been deducted?
  4. In the section 5, the authors say we give some examples, but there is only one example?
  5. section 6, inclusion? or conclusion?
  6. the readers will not be able to understand the applicability of remark 5.2.

Author Response

refer to the attached file  Revision Note of axioms-1398587 Revised.

Reviewer 2 Report

The authors study a nonlinear boundary value problem for interval-valued functions with specific derivative. There are interesting facts in the paper including some surprising results on non-uniqueness of a solution. Also they present an example  illustrating the theoretical conclusions.

The paper can be recommended for publishing after correcting English grammar. For example,

p.2 line 12 from top

it is well known

p.2 line 9 from bottom

can be found in [13]

Section 6

Conclusion

I think the authors should verify the paper once again and improve the English.

Author Response

(The authors gave the same response as above.)

Round 2

Reviewer 1 Report

The authors have incorporated the suggested changes. This version is now okay for publication

Thank you